# Suppressive Effect of *Arctium lappa* L. Leaves on Retinal Damage against A2E-Induced ARPE-19 Cells and Mice

**DOI:** 10.3390/molecules25071737

**Published:** 2020-04-09

**Authors:** Dong Hee Kim, Yae Rim Choi, Jaewon Shim, Yun-Sang Choi, Yun Tai Kim, Mina Kyungmin Kim, Min Jung Kim

**Affiliations:** 1Research Division of Food Functionality, Korea Food Research Institute, Wanju 55365, Korea; donghey543@naver.com (D.H.K.); uiu7895@naver.com (Y.R.C.); jwshim@kfri.re.kr (J.S.); ytkim@kfri.re.kr (Y.T.K.); 2Department of Food Science and Human Nutrition, Jeonbuk National University, Jeonju-si 54896, Korea; minakim@jbnu.ac.kr; 3Department of Food Science and Engineering, Ewha Womans University, Seoul 03760, Korea; 4Research Division of Strategic Food Technology, Korea Food Research Institute, Wanju 55365, Korea; kcys0517@kfri.re.kr; 5Department of Food Biotechnology, Korea University of Science & Technology, Daejeon 34113, Korea

**Keywords:** *Arctium lappa* L. leaves, age-related macular degeneration, A2E accumulation, A2E-induced cell death, apoptosis

## Abstract

Age-related macular degeneration (AMD) is a major cause of irreversible loss of vision with 80–90% of patients demonstrating dry type AMD. Dry AMD could possibly be prevented by polyphenol-rich medicinal foods by the inhibition of N-retinylidene-N-retinylethanolamine (A2E)-induced oxidative stress and cell damage. *Arctium lappa* L. (AL) leaves are medicinal and have antioxidant activity. The purpose of this study was to elucidate the protective effects of the extract of AL leaves (ALE) on dry AMD models, including in vitro A2E-induced damage in ARPE-19 cells, a human retinal pigment epithelial cell line, and in vivo light-induced retinal damage in BALB/c mice. According to the total phenolic contents (TPCs), total flavonoid contents (TFCs) and antioxidant activities, ALE was rich in polyphenols and had antioxidant efficacies on 2,2-diphenyl-1-picrylhydrazyl (DPPH), 2,2′-azino-bis(3-ethylbenzothiazoline-6-sulfonic acid) (ABTS), ferric reducing antioxidant power (FRAP), and 2′,7′-dichlorofluorescin diacetate (DCFDA) assays. The effects of ALE on A2E accumulation and A2E-induced cell death were also monitored. Despite continued exposure to A2E (10 μM), ALE attenuated A2E accumulation in APRE-19 cells with levels similar to lutein. A2E-induced cell death at high concentration (25 μM) was also suppressed by ALE by inhibiting the apoptotic signaling pathway. Furthermore, ALE could protect the outer nuclear layer (ONL) in the retina from light-induced AMD in BALB/c mice. In conclusion, ALE could be considered a potentially valuable medicinal food for dry AMD.

## 1. Introduction

Age-related macular degeneration (AMD) is a degenerative visual disorder of the macula of the retina that affects the central vision of people aged 55 years and above in developed countries and is the leading cause of blindness. It occurs in about 8.7% people worldwide and is expected to increase 1.5 times by 2040 as average life expectancy increases [1]. AMD can be classified into dry and wet forms. Dry AMD is characterized by the formation of drusen deposits between the retinal pigment epithelium (RPE) and the Bruch’s membrane. This is a sign of early dry AMD. Increasing symptoms gradually result in geographic atrophy (medium dry AMD), and vision decreases slowly over the years due to loss of RPE cells and photoreceptors. Dry AMD accounts for 80–90% of AMD patients, and those with geographic atrophy progress to late AMD (wet AMD) [2]. Since patients with dry AMD do not have any early symptoms, self-awareness and prevention constitute the best management.

The main cause of dry AMD has not been identified, but some factors, such as age, smoking, hypertension, obesity, N-retinylidene-N-retinylethanolamine (A2E) accumulation, and blue light illumination, have been hypothesized to be the cause [3,4,5,6]. Among these, only A2E is an intrinsic cause. A2E is abnormally produced in the vitamin A visual cycle and is the main element of lipofuscin, one of the components of drusen [7]. Youth and adults can eliminate A2E from human RPE cells, whereas those over 50–60 years accumulate A2E in RPE due to failure of A2E removal [8]. Continuous A2E accumulation in the RPE cells triggers drusen formation in the macula where the optic nerve and optic cells gather, resulting in DNA damage in RPE cells, RPE cell death via apoptosis, and vision loss [9,10,11]. Therefore, inhibition of A2E accumulation and A2E-induced cell death can play an important role in preventing dry AMD and maintaining RPE function.

*Arctium lappa* L. (AL) is a perennial plant known as edible burdock belonging to the Asteraceae family and is distributed mainly in Asia and Europe. In Korea, AL is popularly used in food and also as traditional herbal medicine as anti-inflammatory, antipyretic, diuretic, and detoxifying agents [12,13]. It also has diverse biological activities, including anti-inflammatory, anti-cancer, antioxidant, neuroprotective, anti-hepatotoxic, anti-diabetic, anti-microbial, and anti-viral effects [14,15,16,17]. Most of these efficacies belong to the roots, seeds, and fruits of AL. The leaves show antimicrobial, anticancer, and antioxidant effects [18,19,20,21,22,23]. AL leaves contain a number of polyphenols, including phenolic compounds and flavonoids, that can attenuate oxidative stress. Since A2E produces oxidative stress, it is possible that the AL leaves inhibit A2E-induced damage to RPE cells, protect the retina, and prevent dry AMD.

Therefore, the purpose of this study was to investigate the protective effects of the extracts of AL leaves (ALE) on A2E accumulation and A2E-induced cell death in ARPE-19 cell, a human retinal pigment epithelial cell line, as well as to elucidate the possible mechanisms of anti-cell death. In addition, the in vivo protective effects of the extracts of AL leaves were monitored on dry AMD animal models using light source exposure to BALB/c male mice.

## 2. Results

### 2.1. Extraction of AL, Total Phenolic Content (TPC), and Total Flavonoid Content (TFC) of ALE

The extraction yield of AL was calculated on the basis of the weight of freeze-dried AL leaves (Figure 1A). Extracting from 10 g of AL with 100% EtOH using ultrasonication yielded 1.35 ± 0.02 g of ALE (135 ± 2 mg/g AL).

The TPC and TFC in ALE were analyzed at different concentrations of ALE (Figure 1B,C). The TPC of ALE was 35.79 ± 1.4, 48.95 ± 7.94, 51.2 ± 3.34, 66.49 ± 1.6, and 101.03 ± 1.66 mg gallic acid equivalent (GAE)/g ALE at 5, 10, 30, 50, 100 μg/mL concentration, respectively. The TFC of ALE was 1.96 ± 0.27, 19.81 ± 0.13, 22.93 ± 0.72, and 41.56 ± 0.3 mg Quercetin equivalent (QE)/g ALE at 10, 30, 50, and 100 μg/mL concentration, respectively. Both TPC and TFC significantly increased in a dose-dependent manner.

### 2.2. Antioxidant Activities of ALE

Antioxidant activities of ALE were monitored using 2,2-diphenyl-1-picrylhydrazyl (DPPH), ABTS, ferric reducing antioxidant power (FRAP), and 2‘,7’-dichlorofluorescin diacetate assays (DCFDA) (Figure 2). DPPH radical scavenging activity of vitamin C (30 μg/mL) was 91.85 ± 0.06% and that of ALE was 3.08 ± 1.68%, 10.94 ± 0.75, and 22.83 ± 3.41% at 30, 50, and 100 μg/mL concentration, respectively. DPPH radical scavenging activity of ALE significantly increased in a concentration-dependent manner.

ABTS radical scavenging activity of vitamin C was 93.48 ± 0.25%, and that of ALE was 4.53 ± 0.21%, 7.53 ± 0.49%, 17.39 ± 0.47%, 25.49 ± 1.09%, and 56.06 ± 0.52% at 5, 10, 30, 50, and 100 μg/mL concentration, respectively. Thus, the ABTS radical scavenging activity of ALE significantly increased in a dose-dependent manner.

According to the FRAP assay, the quantified value of vitamin C was 0.71 mM Fe(II)/g vitamin C and that of ALE was 0.09, 0.13, 0.28, 0.35, and 0.64 mM Fe(II)/g ALE at 5, 10, 30, 50, and 100 μg/mL concentration, respectively. The quantification of FRAP assay with ALE also significantly increased from 30 to 100 μg/mL.

According to DCFDA assay, intracellular reactive oxygen species (ROS) levels were successfully suppressed by ALE. A2E treatment significantly increased intracellular ROS level, whereas ALE pretreatment significantly attenuated A2E-induced intracellular ROS generation in a dose-dependent manner.

### 2.3. A2E Synthesis and Optimization of A2E Concentration

Synthesized A2E was separated by silica gel and purified to a single peak using an HPLC automated fraction collector system. Then, the chromatogram and absorbance spectrum of the purified A2E were confirmed using HPLC and spectrophotometer. The known A2E structure is shown in Figure 3A. A single peak was observed in the chromatogram of the purified A2E detected at 430 nm (Figure 3B). Two peaks with λmax = 435 and 345 nm were visible in the UV spectrum of HPLC-purified A2E (Figure 3C).

To optimize A2E concentration for A2E-induced cell death and A2E accumulation in ARPE-19 cells, ARPE-19 cells were incubated with two concentrations of A2E (10 and 25 μM) for 0.5, 1, 3, 6, 12, or 24 h. A2E at 10 μM had no effects on ARPE-19 cell viability at any time point (Figure 3D). On the other hand, cell viability of ARPE-19 cells treated with 25 μM A2E showed a time-dependent decrease from 3 to 24 h (***p* < 0.01, ****p* < 0.001) (Figure 3E). The reduction rates were 20, 60, 80, and 80% at 3, 6, 12, and 24 h, respectively. Therefore, a lower concentration (10 μM) was used to estimate the inhibition of A2E accumulation by ALE and a higher concentration (25 μM) was chosen to investigate the suppression of A2E-induced cell death by ALE.

### 2.4. Inhibition of A2E Accumulation by ALE

The cytotoxicity of the ALE was evaluated and the effects of ALE on A2E accumulation in ARPE-19 cells were analyzed. Lutein was used as a positive control. ARPE-19 cells were treated with ALE (5, 10, 30 μg/mL) or lutein (5, 10, 30 μg/mL) for 24 h, and then cell viability was confirmed by using the CCK-8 kit. It was found that lutein and ALE did not affect cell viability at all concentrations (Figure 4A). Then, the effects of ALE pretreatment on A2E accumulation in ARPE-19 cells was investigated by measure intracellular A2E concentration. As shown in Figure 4B, ALE attenuated A2E accumulation in a concentration-dependent manner and significantly reduced 40% of A2E accumulation compared to the A2E-treated control at 30 μg/mL (***p* < 0.01). Lutein also inhibited A2E accumulation in a dose-dependent manner. Lutein at 10 and 30 μg/mL significantly suppressed A2E accumulation, demonstrating a 50% reduction at 30 μg/mL lutein (**p* < 0.05, ****p* < 0.001).

### 2.5. Protective Effect of ALE on A2E-Induced ARPE-19 Cell Death by Inhibiting the Apoptotic Signaling Pathway

ARPE-19 cells were treated with 25 μM A2E for 24 h with or without pretreatment with ALE (5, 10, 30 μg/mL) for 24 h, and the protective effect of ALE against A2E-induced cell death was monitored using 3-(4,5-dimethylthiazol-2-yl)-2,5-diphenyltetrazolium bromide (MTT) assay (Figure 4C). Cell viability of ARPE-19 cells was significantly reduced by more than 60% by A2E treatment (****p* < 0.001). ALE showed a slight increase in cell viability at 5 and 10 μg/mL but a significant increase in cell viability (around 20% at 30 μg/mL) compared to A2E-treated control (****p* < 0.001). Lutein increased cell viability in a concentration-dependent manner but significantly increased to about 20% at 30 μg/mL. The protective effect of ALE against A2E-induced cell death was similar to that of lutein.

Apoptosis is one of the ARPE-19 cell death induced by A2E, so changes in apoptosis-associated protein expression were monitored using Western blot analysis (Figure 4D). A2E treatment (25 μM) to ARPE-19 cells caused apoptosis, so protein levels of Bax and cleaved caspase 3 were upregulated, and that of Bcl were downregulated, compared to the control. However, ALE treatment to ARPE-19 cell prior to A2E suppressed apoptosis. The protein levels of Bax and cleaved caspase 3 in ALE-treated cells were lower than in ALE-untreated cells, and that of Bcl-2 in ALE-treated cells was higher than in ALE-untreated cells in a dose-dependent manner. The positive control, lutein, also demonstrated the same effects as ALE.

### 2.6. Inhibitory Effect of ALE on Retinal Damage Caused by White Light

To investigate the effect of ALE on the histological damage of the retina by white light, hematoxylin and eosin (H&E) staining was done (Figure 5A). Light + vehicle exposure group demonstrated histological disturbance of outer segments and inner segments (OS/IS), outer nuclear layer (ONL), inner nuclear layer (INL), and ganglion cell layer (GCL) compared to the control group. The lutein-treated group, a positive control group, moderated the histological disturbances of OS/IS, ONL, INL, and GCL layers compared to the vehicle group despite exposure to light. Similarly, the ALE treatment group mitigates histological disturbances in OS/IS, ONL, INL, and GCL layers depending on the concentration. Figure 5B, which depicts quantified values of these histological disturbances, shows that ALE and lutein significantly inhibited light-induced retinal damage in a dose-dependent manner. According to this result, ALE could protect retinal damage from white light.

## 3. Discussion

This study showed that: (1) ALE contains a large amount of polyphenols, including phenolic compounds and flavonoids, and shows antioxidant effects; (2) A2E accumulation is significantly suppressed by ALE in a concentration-dependent manner; (3) ALE protects ARPE-19 cells against A2E in a concentration-dependent manner by inhibiting the apoptosis signaling pathway; and (4) ALE successfully attenuates retinal damage using a light-induced animal model of dry AMD. These findings indicated that ALE could be functional food/ medicines for the prevention of AMD development and progression.

AMD is a disease that causes blindness due to the damage and death of RPE cells and photoreceptor cells [24]. The molecular hallmark of AMD, especially dry AMD, is the presence and accumulation of drusen within RPE cells or in the interface between RPE cells and Bruch’s membrane [25]. Drusen contains diverse components, including lipofuscin, complement components, apolipoprotein E, amyloid P component, and complement factor H [25,26,27,28,29]. Most drusen is lipofuscin, and more than 90% of the lipofuscin was generated from the visual retinoid cycle and was resistant to degradation by lysosomes and proteasomes from RPE cells with aging. A major component of lipofuscin from the RPE cells is a fluorophore A2E [30]. A2E is a cytotoxic lipofuscin bis-retinoid, formed from the reaction between all-*trans*-retinal and ethanolamine in the photoreceptor cells, and is accumulated in the RPE cells [31]. A2E-laden cells increase chronic oxidative stress and excessive A2E leads to cell death [32,33]. A2E can be amassed in human RPE cells in vivo up to 60–130 ng/10^5^ cells, and does not cause significant damage to DNA up to 20 μM [34,35]. In addition, 10 μM A2E has no effect on the cell viability of ARPE-19 cells for 48 h, whereas 25 and 50 μM A2E trigger cell death in a time dependent manner [36,37]. We also described the same results as the effect of A2E (10 and 25 μM) on ARPE-19 cell viability that can reflect clinical phenomena for lipofuscin accumulation and damage over a lifespan.

Lipofuscin, such as A2E, can be accumulated in lysosomes and cytosol. Lysosomal lipofuscin triggers not only protein oxidation, aggregation, and lipofuscin production but also the induction of oxidative stress by destructive contributions of autophagy and lysosomal degradation. Cytosolic lipofuscin also generates ROS. Therefore, cytosolic protein degradation suffers, and cell viability decreases. With age, lipofuscin aggregates in RPE cells, producing more oxidative stress. A2E produces free radicals in organized media and produces radicals, such as superoxide and peroxyl [38,39]. Therefore, suppressing oxidative stress will be valuable in preventing AMD.

Plant-based foods contain a large amount of polyphenols with antioxidant properties [40]. ALE also consist of phenolic compounds and flavonoids and showed antioxidant efficacy. According to antioxidant assays, ALE significantly cause DPPH radical, ABTS radical and ferrous ion (Fe^2+^) scavenging activity and significantly decrease A2E-induced intracellular ROS generation. Phenolic compounds and flavonoids prevent oxidative stress-associated retinal disease. AL leaves also contain polyphenols, such as chlorogenic acid, 1, 5-dicaffeoylquinic acid, caffeic acid, arctiin, rutin, luteolin, and quercetin [19,20]. Most of these showed antioxidant effects according to DPPH, ABTS, and FRAP assays. Furthermore, some compounds, such as chlorogenic acid and quercetin, can defend RPE cells from light and oxidative damage using cell-based assays, which work by inhibiting apoptotic pathways [41,42]. Natural antioxidants, such as zeaxanthin, lutein, bilberry, and polyphenol compounds, have protective effects on A2E-induced APRE-19 cell death [43,44]. In addition, the extracts of polyphenol-rich *Vaccinium uliginosum* L. or *Prunella vulgaris* var. L. inhibit blue light-induced apoptosis in cells containing A2E in vitro and in vivo [45,46,47]. Therefore, ALE could affect APRE-19 cell viability against A2E.

ALE protected ARPE-19 cells against A2E by two means: regulation of intracellular A2E accumulation, and modulation of A2E-laden cell death by regulating the apoptotic signaling pathway. Despite the continuous A2E exposure (10 μM), ALE inhibited the increase of intracellular A2E concentration in a dose-dependent manner with levels similar to lutein, compared with A2E-treated control. In addition, ALE successfully suppressed ARPE-19 cell death against A2E, similar to lutein. The mechanism of lipofuscin accumulation in RPE cells is not exactly known, but is believed to be proteostasis. Aging attenuates cellular proteolysis. Consequently, unremoved misfolded proteins form complexes with perinuclear/centrosomal-proximal proteins, which produce aggresomes [48,49,50,51,52]. Lipofuscin is one of these aggresomes and is located within lysosomes via macroautophagy, and in cytosol [53]. Accumulated lipofuscin causes a vicious cycle. The large amount of lipofuscin suppresses proteasomes, impedes the breakdown of oxidized proteins, increases ROS generation, and then induces additional lipofuscin formation [54,55,56]. Finally, lipofuscin causes cytotoxicity in RPE cells. Lipofuscin-laden RPE cells cause dysfunction and undergo apoptosis via the intrinsic pathway [57]. During intrinsic apoptosis, anti-apoptotic proteins, such as Bcl-2, are deactivated and pro-apoptotic protein, such as Bax, are activated. This promotes the release of cytochrome C and formation of cleaved caspase 3. This study showed that ALE suppressed A2E-induced apoptosis by the upregulation of Bcl-2 and downregulation of Bax and cleaved caspase 3. Therefore, ALE could potentially prevent and inhibit A2E-mediated dry AMD by the inhibition of oxidative stress, A2E accumulation, and apoptosis.

The protective effect of ALE shown in vitro was correlated with an in vivo animal model. BALB/c mice were exposed to high power white light (10,000 lux). The light source did not reduce the thickness of the ONL layer, as in previous research, because we sacrificed the mice immediately after light exposure. However, histological disturbances in the OS/IS, ONL, INL, and GNL layers were observed when compared with the control. This phenomenon hass also appeared in other research, like that of Shibagaki et al. (2015), who showed the destruction of IS/OS and irregular ONL [58]. The density between the nuclei in the retina in ALE- or lutein-treated group was higher than in the light-treated group. ALE protected the retina in a dose-dependent manner, and the efficacy of 200 mg/kg ALE was similar to that of 50 mg/kg lutein. For the AMD in vivo study, a single animal model could not be used as it would not represent all the signs of AMD [59]. However, a common phenomenon exists in all animal models. The common features of AMD are generation of oxidative stress, loss of ONL thickness, and a histological disturbance in the whole layer [60]. Rodent retina is composed of OS/IS, ONL, INL, and GNL layers. After AMD induction, intracellular organelles, such as DNA, are damaged by the generated ROS. The number of nuclei in the retinal layers is reduced and most retinal layers, especially the ONL layer, become thin. Monitoring these changes may contribute to understanding the effects of medicinal foods or medicines. Many research studies have shown that light sources (white or blue) cause apoptosis in the photoreceptors of animal models [45,46,47,61]. This light-inducing AMD animal model was used for the discovery of AMD-protected medicinal plants or foods, such as the extracts of *Prunella vulgaris* var. L., *Curcuma longa* L., and *Vaccinium uliginosum* L. These research studies observed H&E stained retinal images and analyzed changes in the thickness of the whole retinal layer or ONL layer. The extracts effectively protect the retinal layer or ONL layer against blue light. Lutein, a well-known medicine for AMD treatment, also inhibits light-induced damage in retina similarly.

ALE had antioxidant efficacy and appeared to protect ARPE-19 cells by inhibiting A2E accumulation and apoptosis against A2E. ALE also suppressed light-induced retinal damage in BALB/c mice. Considering human equivalent dose (HED) and extraction yield of ALE, taking the extract is more effective for preventing dry AMD than taking raw ingredients. Equivalent doses of ALE at 50, 100, and 200 mg/kg BW in mice are 4.05, 8.1, and 16.2 mg/kg BW in human, and the serving size for 16.2 mg ALE/kg BW is 120 mg AL/kg BW due to extraction yield (135 mg/g AL). Thus, ALE could be a potentially preventive medicinal food for dry AMD.

## 4. Materials and Methods

### 4.1. Reagents

AL leaves were purchased from Jirisan Starmaru (Sancheong, Korea). Lutein powder (40%) was received as a present from the Novarex Co., Ltd. (Seoul, Korea). Sodium hydroxide (NaOH), sodium nitrite (NaNO_2_), aluminium (III) chloride (AlCl_3_), and sodium carbonate (Na_2_CO_3_) were acquired from Junsei Chemicals (Tokyo, Japan). All-trans-retinal, triton X-100, ethanolamine, acetic acid, ascorbic acid, gallic acid, 2,2-diphenyl-1-picrylhydrazyl (DPPH), ABTS tablets, ABTS buffer, 2,4,6-tripyridyl-S-triazine (TPTZ), quercetin, potassium persulfate, iron(III) chloride (FeCl_3_), dimethyl sulfoxide (DMSO), chloroform, and Folin-Ciocalteau reagent were obtained from Sigma-Aldrich (St. Louis, MO, USA). Ethanol (EtOH) was purchased from Merck (Darmstadt, Germany). Methanol (MeOH), acetonitrile, and water were procured from JT BAKER Chemical Co., (Phillipsburg, NJ, USA). Trifluoroacetic acid (TFA) and sodium acetate were purchased from Thermo Fisher Scientific (Waltham, MA, USA). 1N Hydrochloric acid was obtained from DAESUNG Co, Ltd. (Siheng, Korea). Polyclonal anti-cleaved-caspase-3, anti-caspase-3, anti-Bcl-2, and anti-Bax antibodies were purchased from Cell Signaling Technology Inc. (Danvers, MA, USA). A stock solution of lutein was prepared at a concentration of 10 mM in DMSO. DMSO in the culture media was less than 0.1%.

### 4.2. Ethanolic Extraction of AL Leaves

Fresh AL leaves were washed with flowing water, cut to a suitable size, and lyophilized. Freeze-dried AL (10 g) was added to ethanol (500 mL), and the mixture was extracted by ultrasonication using an ultrasonic processor (VCX 750; Sonics & Materials, Newtown, CT, USA) for 2 h. The treatment conditions were as follows: 750 W power output, 20 kHz frequency, 80% amplitude, and 03 s/ 03 s pulses. The temperature was maintained at 4 °C during ultrasonication using a low-temperature bath (NCB-2200; EYELA Co., Tokyo, Japan). ALE was concentrated using a rotary evaporator (EYELA Co., Tokyo, Japan), freeze-dried, and stored at –80 °C. For cell treatment, ALE was dissolved in DMSO to yield 100 mg/mL stocks and ALE stock solution (100 mg/mL) was diluted at each concentration in a culture medium. The final concentration of DMSO in media was less than 0.1%.

### 4.3. Synthesis of A2E

A2E was synthesized from all-trans-retinal and ethanolamine [35]. Mixtures of all-trans-retinal, ethanolamine, and acetic acid in ethanol were stirred in the dark, at room temperature, for 72 h. The mixture was purified by silica gel column chromatography. The mixture was loaded in a silica gel column that was prewashed with a mixed solvent (methanol: dichloromethane = 5:95). Then, A2E was eluted with a mixed solvent (methanol: dichloromethane = 8:92) containing 0.02% TFA. After separation, A2E was concentrated under nitrogen gas in a concentrator and purified to a single peak using Ultimate 3000 HPLC system (Dionex Corporation, Sunnyvale, CA, USA) with an automated fraction collector system (AFC-3000 UltiMate Fraction Collector; Thermo-Fisher Scientific, Waltham, MA, USA). The purified A2E was confirmed by a chromatogram and absorbance spectrum using HPLC and spectrophotometer, respectively. A2E stock solution was 10 mM in DMSO and was stored at –20 °C in the dark. In each experiment, A2E stock solution was diluted in a culture media at concentration of 10 and 25 μM.

### 4.4. Cell Culture

ARPE-19, a human RPE cell from American Type Culture Collection (ATCC; Manassas, VA, USA), was grown in Dulbecco’s modified Eagle’s medium Nutrient Mixture F-12 (DMEM/F12, Gibco, Gaithersburg, MD, USA) supplemented with 10% heat-inactivated fetal bovine serum (FBS, Gibco, Gaithersburg, MD, USA) and 1% penicillin/streptomycin (Gibco, Gaithersburg, MD, USA) at 37 °C in 5% CO_2_. When ARPE-19 cells were grown to 70–80% confluence on culture dishes, cells were seeded on 96-well or 24-well microplates (2 × 10^4^ cells/well or 1 × 10^5^ cells/well, respectively) and used in all experiments.

### 4.5. Determination of Total Phenolic Contents (TPCs)

TPCs were measured by modifying the Folin-Ciocalteu method [62]. The extract (125 μL) was diluted at each concentration with MeOH and mixed with 1 N Folin-Ciocalteau reagent (375 μL) in a 1.5 mL tube, at room temperature, for 5 min. After 5 min, 700 mM sodium carbonate solution (500 μL) was added thereto, mixed, and allowed to react for 1 h. Then, the absorbance of the supernatant was measured at 765 nm using a spectrophotometer (SpectraMax M2e, Molecular Devices, Sunnyvale, CA, USA). TPCs were calculated from a calibration curve with gallic acid and depicted as milligrams of gallic acid equivalent (GAE) per gram of dry weight of ALE (mg GAE/g ALE). The calibration curve for GAE was obtained using 0.00625, 0.0125, 0.025, 0.05, and 0.1 mg/mL gallic acid in 100% MeOH.

### 4.6. Determination of Total Flavonoid Contents (TFCs)

TFC was determined according to the aluminum chloride colorimetric method [63]. The extracts in MeOH were sequentially reacted with 5% NaNO_2_ for 6 min, 10% AlCl_3_ for 5 min, and 1 M NaOH, for 15 min, at room temperature. The absorbance of the mixture was measured at 492 nm with a spectrophotometer. TFCs were calculated from a calibration curve with quercetin and were expressed in terms of milligrams of Quercetin equivalent (QE) per gram of dry weight of ALE (mg QE/g ALE). The standard curve for QE was obtained using 0.00625, 0.0125, 0.025, 0.05, and 0.1 mg/mL of quercetin in 100% MeOH.

### 4.7. Determination of Antioxidant Capacity

#### 4.7.1. DPPH Radical Scavenging Assay

The extracts in MeOH was mixed with 0.4 mM DPPH solution and incubated in the dark, at room temperature, for 30 min. The absorbance of the reaction solution was read at 517 nm. Ascorbic acid (30 μg/mL) was used as a positive control. DPPH radical scavenging activity was expressed as percentage according to the following equation.
(1)$$\mathrm{DPPH}\;\mathrm{radical}\;\mathrm{scavenging}\;\mathrm{activity}\;(\%)=\left(\frac{Abs\;Control-Abs\;Sample}{Abs\;Control}\right)*100.$$

#### 4.7.2. ABTS Radical Scavenging Assay

A mixture of 2.45 mM potassium persulfate solution and 7 mM ABTS solution in a 1:1 ratio was reacted for 14–16 h in the darkroom to produce ABTS radical cation (ABTS+•). Then, ABTS+• solution with 0.70 ± 0.02 absorbance at 734 nm was obtained by dilution in MeOH. The extracts in MeOH and ABTS+• solution in a 1:1 ratio were reacted in the dark, at room temperature, for 10 min. The absorbance of the reaction solution was measured at 734 nm. Ascorbic acid (30 μg/mL) was used as a positive control. ABTS radical scavenging activity was expressed as percentage according to the following equation:(2)$$\mathrm{ABTS}\;\mathrm{radical}\;\mathrm{scavenging}\;\mathrm{activity}\;(\%)=\left(\frac{Abs\;Control-Abs\;Sample}{Abs\;Control}\right)*100.$$

#### 4.7.3. Ferric Reducing Antioxidant Power (FRAP) Assay

Acetate buffer (0.3 M, pH 3.6) in D.W, TPTZ solution (10 mM) in 40 mM HCl, and FeCl_3_ solution (20 mM) in D.W were prepared. Then, acetate buffer, TPTZ solution, and FeCl_3_ solution were mixed in a ratio of 10:1:1 and allowed to react at 37 °C for 10 min to make the FRAP reagent. The extracts and FRAP reagent were mixed in a ratio of 1:10 at 37 °C for 10 min, and absorbance was measured at 593 nm. The FRAP value was calculated using the calibration curve with Trolox and expressed as mM of ferrous ion (Fe^2+^) per gram of ALE. Ascorbic acid (30 μg/mL) was used as a positive control.

#### 4.7.4. 2‘,7’-Dichlorofluorescin Diacetate (DCFDA) Assay

Generated intracellular ROS was detected using DCFDA assay. ARPE-19 cells in 96-well plates were treated with the extracts at 5, 10, and 30 μg/mL for 1 h, washed with phosphate-buffered saline (PBS), and then, treated with 25 μM A2E for 1 h. As a control, ARPE-19 cells were pre-incubated with 0.1% DMSO for 1 h and then with or without A2E (25 μM) for 1 h. Lutein (30 μg/mL) was used as a positive control. During the last 30 min, cells were stained with 5-(and-6)-chloromethyl-2′,7′-dichlorodihydrofluorescein diacetate, acetyl ester (CM-H_2_DCFDA, 10 μM; Thermo Fisher Scientific, Waltham, MA, USA) in an incubator. Then, the cells were washed with PBS and the Relative Fluorescent Unit (RFU) in the cells was monitored using a spectrophotometer at excitation and emission wavelengths of 488 and 525 nm, respectively. All experiments were carried out in the dark to diminish photo-oxidation and photo-reduction.

### 4.8. Cell Viability Assay

To optimize A2E concentration, the cell viability was determined using cell counting kit-8 (CCK-8, Enzo Life Science, Farmingdale, NY, USA). ARPE-19 cells in 96-well microplate were treated with 10 μM and 25 μM of A2E solution. At each time point (0, 0.5, 1, 3, 6, 12, and 24 h), A2E-containing culture medium was replaced to 100 μL of culture media containing 10 μL of CCK-8 solution in each well and further incubation was carried out for 2 h at 37 °C. The absorbance was measured at 450 nm using a spectrophotometer and normalized as follows:Cell viability = absorbance at each time point/ absorbance at 0 h.(3)

Cytotoxicity of ALE was estimated using CCK-8. ARPE-19 cells in 96-well were treated with ALE (5, 10, and 30 μg/mL) or lutein (5, 10, and 30 μg/mL) for 24 h, washed with PBS, and incubated with CCK-8 reagent for 2 h. The absorbance was measured at 450 nm using a spectrophotometer. The absorbance of each experimental group was normalized to absorbance of 0.1% DMSO-treated control group.

The MTT cell viability assay was used to monitor the effect of A2E on ALE-pretreated ARPE-19 cells. ARPE-19 cells pretreated with ALE (5, 10, and 30 μg/mL) or lutein (5, 10, and 30 μg/mL) for 24 h were washed with PBS and incubated with A2E (25 μM) for 24 h. After 24 h, MTT assay was carried out and the absorbance was measured at 570 nm using a spectrophotometer. The absorbance was normalized as follows:Cell viability = absorbance of each experimental group/absorbance of control group.(4)

### 4.9. Intracellular A2E Accumulation

Accumulated intracellular A2E is measured by the method depicted in Figure 6A. ARPE-19 cells (1 × 10^5^ cells/well) were seeded in 24-well plates on 1st day. Thereafter, ARPE-19 cells were treated with ALE or lutein at 5, 10, and 30 μg/mL on 2nd, 5th, and 8th days and with A2E (10 μM) on the 3rd, 6th, and 9th day. Washing with PBS was performed between sample and A2E treatment. On the last day (10th day), ARPE-19 cells were washed with PBS, scraped and harvested in the presence of 0.5% triton X-100. Collected cells were sonicated for 1 min to destroy the layers. Some of cell lysates were used to measure protein levels using BCA assay, and most were used to quantify accumulated A2E as follows; Cell lysate was extracted three times with chloroform added each time. Collected fraction containing A2E in chloroform was filtered using polytetrafluoroethylene (PTFE) filter, evaporated under nitrogen gas, and dissolved in ethanol for HPLC analysis. HPLC system consists of Dionex Ultimate-3000 (Thermo Scientific, Sunnyvale, CA, USA) with a quaternary pump, an autosampler, a UV-Vis diode array detector (DAD3000; Thermo Scientific, Sunnyvale, CA, USA), and a reverse-phase C18 column (4.6 × 250 mm; 5 μm particle size; Agilent Technologies, Santa Clara, CA, USA). The detection wavelength for A2E was set at 340 and 430 nm. The mobile phase comprised of water with 0.1% TFA (solvent A) and acetonitrile with 0.1% TFA (solvent B). A2E was eluted as follows: 85% solvent B in the beginning, increased to 96% in 10 min, maintained for 5 min, increased to 100% solvent B in 17 min, and maintained until 25 min. The flow rate was set to 1 mL/min. The sample injection volume was 10 μL, and all standards and samples were filtered through a 0.2 μm PTFE filter before injection. A2E was quantified using external standard and divided by protein level (A2E concentration/protein value). Then, quantified A2E was normalized as follows:(5)$$Relative\;A2E\;level=\frac{A2E\;\mathrm{amount}\;\mathrm{in}\;\mathrm{each}\;\mathrm{experimental}\;\mathrm{group}}{A2E\;\mathrm{amount}\;\mathrm{in}\;A2E-\mathrm{treated}\;\mathrm{control}\;\mathrm{group}}.$$

### 4.10. Western Blotting

ARPE-19 cells were treated with the extract at 5, 10, and 30 μg/mL for 24 h and with A2E (25 μM) for 24 h. Then, the cells were harvested and lysed with lysis RIPA buffer (150 mM NaCl, 1% Triton X-100, 1% sodium deoxycholate, 0.1% SDS, 50 mM Tris-HCl (pH 7.5), 2 mM EDTA (pH 8.0)) containing protease inhibitor cocktail and phosphatase inhibitor cocktail. The protein obtained was electrophoresed with 12% sodium dodecyl sulfate-polyacrylamide gel electrophoresis (SDS-PAGE) and transferred to a polyvinylidene fluoride (PVDF) membrane, which was blocked with 5% bovine serum albumin (BSA) for 2 h, incubated overnight with the primary antibodies (Bax, Bcl-2, caspase 3, and cleaved caspase 3) at 4 °C, and treated with secondary antibody for 2 h. Protein expressions of Bax, Bcl-2, caspase 3, and cleaved caspase 3 were detected using Western blotting detection enhanced chemiluminescence (ECL) reagent. Expression levels were visualized using a Vilber Fusion Solo S chemiluminescence acquisition system (Vilber Lourmat, France).

### 4.11. Animal Study

Animal experiments were used in accordance with the guidelines of the Animal Control and Use Committee (IACUC) of the Korea Food Research Institute (KFRI-M-19049). BALB/c male mice (4 week-old) were purchased from Orient Bio, Inc. (Seongnam, Korea). Mice were housed in standard cage and maintained at a temperature of 23 ± 2 °C, 50 ± 10% humidity, and periodic illumination (130–350 lux, 12 h light-dark cycle) with free access to standard laboratory chow and tap water *ad libitum.* The scheme of the animal study is shown in Figure 6B. After adaption for 1 week, mice were divided into six different groups and group-housed (*n* = 6 for each group, three mice per cage): control, only light exposure, light + 50 mg/kg body weight (BW)/day of lutein, light + 50 mg/kg BW/day of ALE, light + 100 mg/kg BW/day of ALE, and light + 200 mg/kg BW/day of ALE. ALE dissolved in PBS was orally administered for 4 weeks (5 days/week). The control and only light exposure groups received the vehicle (PBS) for 4 weeks by oral gavage (5 days/week). After 4 weeks, the mice, except those in the control group, were exposed to light with a white lamp (10,000 lux) for 6 h in a light cage with stainless steel walls and bottom. The cage contained a ventilator, and the mice were subjected to dark adaptation for 12 h before light exposure. The control group was maintained in the light box without light.

After 6 h of light exposure, the mice were immediately given inhaled anesthesia with isoflurane and were euthanized to remove the eyes. The extracted eye was fixed in 4% paraformaldehyde (PFA) and then embedded in paraffin wax. The paraffin-embedded retinal section (3 um thickness) was stained with H&E) for histological examination. The stained region in the ONL layer was measured using metamorph (MetaMorph^®^ Image Analysis Software Version 4.0, Universal Imaging Corp. Downingtown, PA, USA) and calculated using the following equation:(6)$$\mathrm{ONL}\;\mathrm{nuclei}\;\mathrm{area}\;(\%)=\left(\frac{Stained\;area\;in\;ONL\;layer}{Total\;area\;of\;ONL}\right)*100.$$

### 4.12. Statistical Analysis

The results are presented as Mean ± Standard Deviation (SD) (*n* ≥ 3). Statistical analysis was performed in the GraphPad Prism 5 software (GraphPad, San Diego, CA, USA). Differences in the mean values between groups were determined using one-way analysis of variance (ANOVA) followed by Tukey’s honestly significant differences (HSD) test.

## 5. Conclusions

The present study provided evidence that ALE protects human ARPE-19 cells and BALB/c mice from A2E-induced and light-induced damages, respectively. ALE containing polyphenols (phenolic compounds and flavonoids) showed antioxidant and protective effects on A2E accumulation and A2E-induced cell death. In addition, ALE inhibited A2E-induced apoptosis by the suppression of caspase 3 cleavage and Bax, as well as the activation of Bcl-2. The inhibitory effect of ALE on dry AMD was confirmed by BALB/c mice. It was seen that the junction between cells in the ONL layer became stable against light exposure. Prevention is best for dry AMD with no symptoms. Therefore, food materials are a good natural source for the prevention of dry AMD. Our study suggests that the leaves of AL could be considered as potential medicine for preventing AMD.

## Figures and Tables

**Figure 1 molecules-25-01737-f001:**
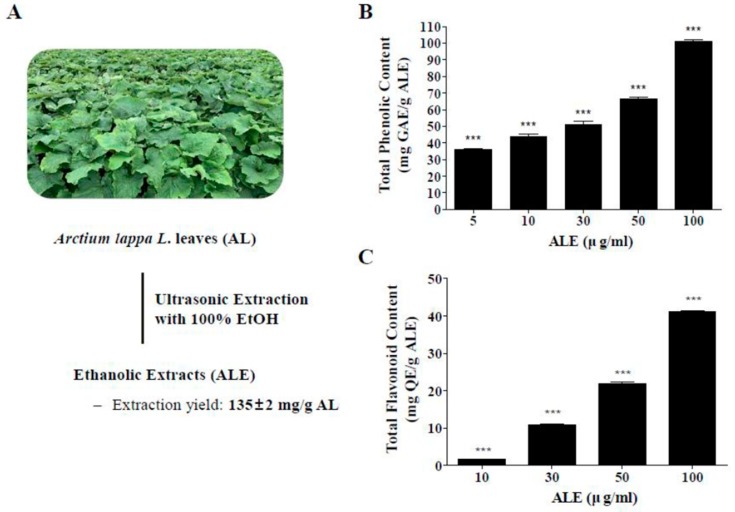
Extraction of AL and total polyphenol contents in extract of AL leaves (ALE). (**A**) Ethanolic extracts and extraction yield of ALE. (**B**) TPCs in ALE in a dose range from 5 to 100 μg/mL. (**C**) TFCs in ALE in a dose range from 10 to 100 μg/mL. The values represent the mean ± SD (*n* ≥ 3). ****p* < 0.001 vs blank group, one-way ANOVA with Tukey’s post hoc test. TPC, total phenolic content; TFC, total flavonoid content; GAE, gallic acid equivalent; QE, quercetin.

**Figure 2 molecules-25-01737-f002:**
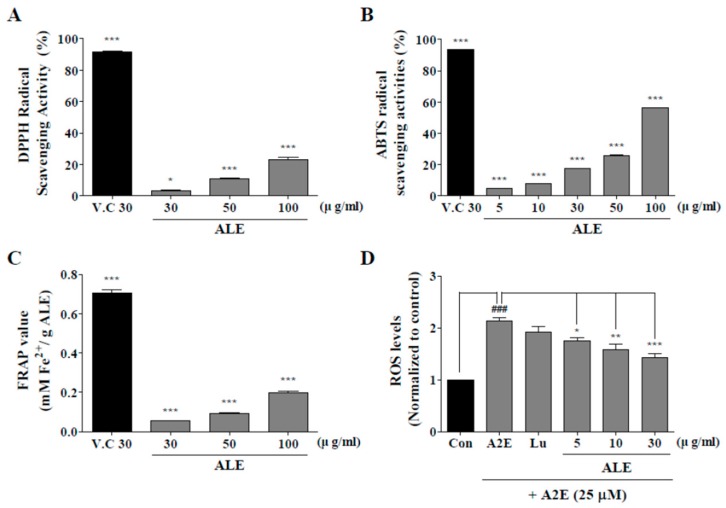
Antioxidant activity of ALE. 2,2-diphenyl-1-picrylhydrazyl (DPPH) radical scavenging assay (**A**), 2,2’-azino-bis(3-ethylbenzothiazoline-6-sulfonic acid) (ABTS) radical scavenging assay (**B**), and ferric reducing antioxidant power (FRAP) assay (**C**) were used to monitor the antioxidant activities of ALE at different doses. (**D**) ARPE-19 cells, a human retinal pigment epithelial cell line, were treated with ALE prior to N-retinylidene-N-retinylethanolamine (A2E) and intracellular reactive oxygen species (ROS) generation was monitored with the use of 5-(and-6)-chloromethyl-2′,7′-dichlorodihydrofluorescein diacetate, acetyl ester (CM-H_2_DCFDA). The values represent the mean ± SD (*n* ≥ 3). **p* < 0.05, ****p* < 0.001 vs blank group, one-way ANOVA with Tukey’s post hoc test.

**Figure 3 molecules-25-01737-f003:**
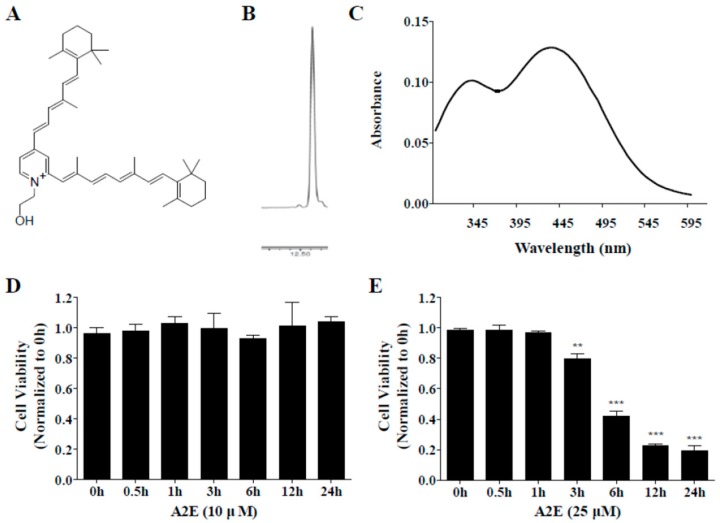
Characteristics of synthesized A2E. (**A**) Chemical structure of A2E. (**B**) Chromatogram of HPLC-purified A2E by HPLC. (**C**) UV spectra and structure of synthesized A2E. (**D**,**E**) Optimization of A2E concentration. ARPE-19 cells were treated with A2E at 10 μM (**D**) or 25 μM (**E**) for 24 h, and cell viability was measured by cell counting kit (CCK)-8 kit at different time points. The values represent the mean ± SD (*n* ≥ 3). ***p* < 0.01, ****p* < 0.001 vs 0 h group, one-way ANOVA with Tukey’s post hoc test.

**Figure 4 molecules-25-01737-f004:**
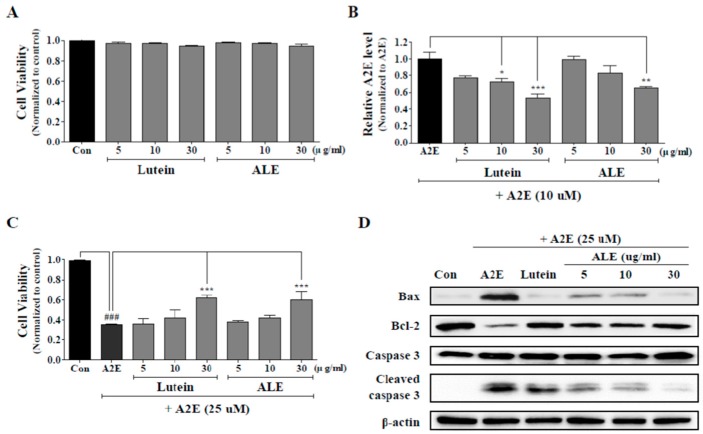
Protective effects of ALE on A2E accumulation and A2E-induced cell death in ARPE-19 cells. (**A**) Cell viability of ARPE-19 cells treated only with lutein or ALE. ARPE-19 cells were incubated with lutein or ALE (5, 10, 30 μg/mL) for 24 h and the cell viability was measured using the CCK-8 kit. (**B**) Inhibition of intracellular A2E accumulation by ALE pretreatment. ARPE-19 cells were treated with ALE (5, 10, and 30 μg/mL) on days 2, 5, and 8, and with A2E (10 μM) on days 3, 6, and 9. On the last day (day 10), A2E in ARPE-19 cells was extracted, analyzed by HPLC, quantified using an external A2E standard curve, and normalized to protein level. The values represent the mean ± SD (*n* ≥ 3). * *p* < 0.05, ** *p* < 0.01, *** *p* < 0.001 vs A2E group, one-way ANOVA with Tukey’s post hoc test. (**C**) Inhibition of A2E-induced cell death by ALE pretreatment. ARPE-19 cells were incubated with lutein or ALE (5, 10, and 30 μg/mL) for 24 h, prior to A2E treatment (25 μM) for 24 h. Then, cell viability was measured by 3-(4,5-dimethylthiazol-2-yl)-2,5-diphenyltetrazolium bromide (MTT) assay. The values represent the mean ± SD (*n* ≥ 3). ^###^
*p* < 0.001 vs C (control) group; *** *p* < 0.001 vs A2E group, one-way ANOVA with Tukey’s post hoc test. (**D**) Protein expression of apoptosis-associated factors in A2E-laden APRE-19 cells.

**Figure 5 molecules-25-01737-f005:**
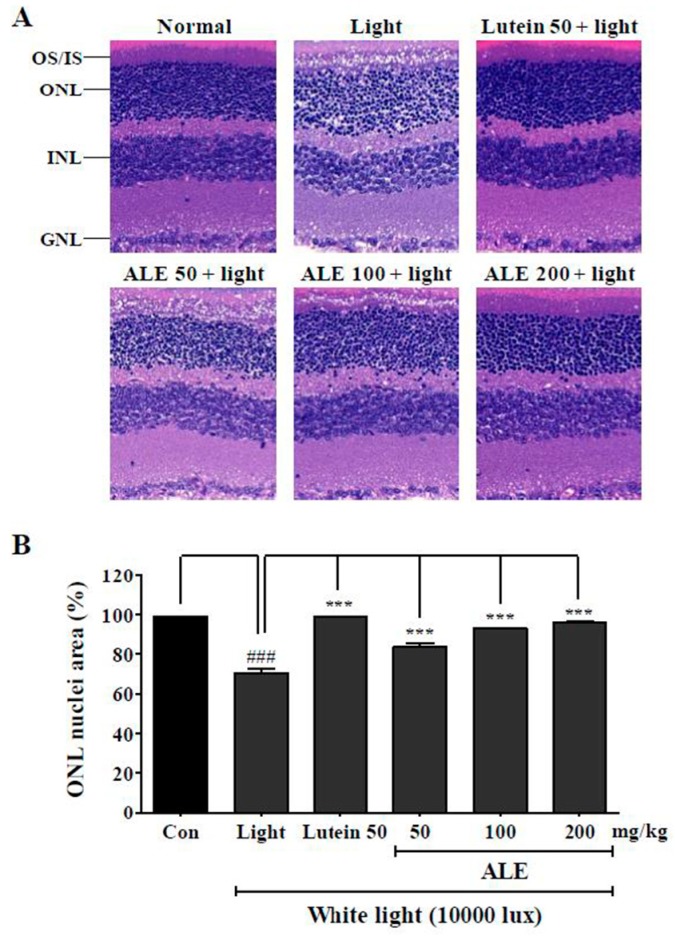
Inhibitory effect of ALE on light-induced retinal damage. (**A**) Representative hematoxylin and eosin (H&E) stained images in light-induced age-related macular degeneration (AMD) model. BALB/c mice (*n* = 6) were treated with vehicle, lutein (50 mg/kg), or ALE (50, 100, and 200 mg/kg) for 4 weeks and exposed to white light at 10,000 lux in light cages for 6 h. After exposure, mice were given inhaled anesthesia with isoflurane immediately and were euthanized to remove the eyes. (**B**) Stained region in ONL layer was calculated using metamorph and expressed in ONL nuclei area (%). The values represent the mean ± SD (*n* = 3). ^###^: *p* < 0.001 vs control group; *** *p* < 0.001 vs light exposure group, one-way ANOVA with Tukey’s post hoc test. OS, outer segments; IS, inner segments; ONL, outer nuclear layer; INL, inner nuclear layer; GCL, ganglion cell layer.

**Figure 6 molecules-25-01737-f006:**
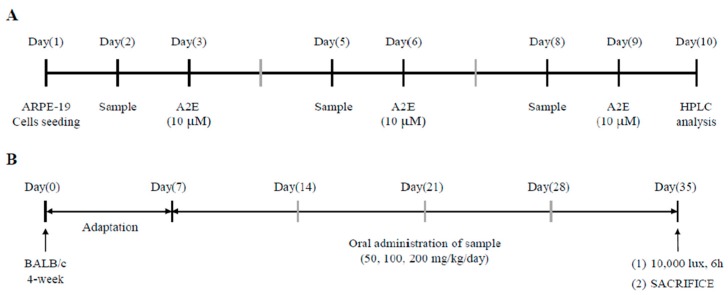
Scheme of treatment schedule. (**A**) Schedule for the inhibitory effect of ALE on A2E accumulation in ARPE-19 cells. (**B**) Schedule for rodent study.

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
