# Peer review of "Suppressive Effect of Arctium Lappa L. Leaves on Retinal Damage Against A2E-Induced ARPE-19 Cells and Mice"

_molecules, 2020, doi:10.3390/molecules25071737_

Round 1

Reviewer 1 Report

The submission of Kim et al. reported the effect of Arctium lappa leaves ethanol extract on retinal damage induced by A2E both using in vitro and in vivo model. The results are of some interest although the paper needs to be improved before publication. Major shortcomings is related to the lack of identification of potential bioactive compounds in the ALE extract.

  • Paragraph 2.2. Why the authors chose ethanol as solvent?
  • Paragraph 2.7.4.2 Where was dissolved the extract? It is not clear if cells were pretreated with ALE or the extract and A2E were added at the same time in the cell culture media. Also it is not clear which type of control assays have been carried out
  • Paragraphs 2.8 and 2.9 needs English revision for the presence of bad structured sentences. Where was dissolved the extract?
  • Paragraphs 2.9 and 2.11 Figure 6 should be renamed as Figure 1.
  • Paragraph 2.10 Which extract concentration? Where was dissolved the extract?
  • The sentence “The quantification of fluorescence intensity of ALE-treated ARPE-19 cells significantly increased from 30 to 100 μg/ml, according to the DCFDA assay” page 7 is not clear. If fluorescent increase means more ROS production. Also in figure the reported data are with ALE concentration of 5-10-30 and not 50 or 100 ug/mL
  • Please reformulate the sentences “There was no significant difference in cell viability from 0 h to 24 h when A2E was treated with 10 μM” page 8 and “ALE and A2E were sequentially treated to ARPE-19 cells and intracellular A2E concentration was analyzed” page 10

Reviewer 2 Report

The authors describe an interesting study of the retinal protection from light through the application or intake of AL. The design and analyses appear sound and the conclusions appropriate. A few comments:

Introduction:

AL (Burdock) is a perennial plant belonging... Define AL

Methods:

2.11 Specify if oral gavage was used. Were mice fed standard chow? Please specify the chow. Were mice group-housed?

Results:

Figure 6 should be changed to Figure 1 and the other figures reordered.

 3.6 Define each retinal layer (OS, IS, ONL ,INL, GCL)

Discussion:

Page 12: "ALE protects ARPE-19 cell death against..." should read, "ALE protects ARPE-19 cells against A2E..."

Add to Discussion: What is an equivalent dose of ALE in humans for each concentration given to mice? For the foods including ALE that you mention in the Introduction,  what serving size would contain the maximum dose you used?

Round 2

Reviewer 1 Report

The authors improved their manuscript. In my opinion is now acceptable for publication